

# Cognitive performance in distinct groups of children undergoing epilepsy surgery—a single-centre experience

Barbora Benova[1,2,*], Anezka Belohlavkova[1,2,*], Petr Jezdik[3], Alena Jahodova[1,2], Martin Kudr[1,2], Vladimir Komarek[1,2], Vilem Novak[1,4], Petr Liby[5], Robert Lesko[1,5], Michal Tichy[1,5], Martin Kyncl[1,6], Josef Zamecnik[1,7], Pavel Krsek[1,2] and Alice Maulisova[8]

[1] 2nd Faculty of Medicine, Charles University Prague, Prague, Czech Republic
[2] Department of Paediatric Neurology, Motol University Hospital, Prague, Czech Republic
[3] Faculty of Electrical Engineering, Department of Circuit Theory, Czech Technical University of Prague, Prague, Czech Republic
[4] Department of Paediatric Neurology, Ostrava Faculty Hospital, Ostrava, Czech Republic
[5] Department of Neurosurgery, Motol University Hospital, Prague, Czech Republic
[6] Department of Radiology, Motol Universiy Hospital, Prague, Czech Republic
[7] Department of Pathology and Molecular Medicine, Motol University Hospital, Prague, Czech Republic
[8] Department of Clinical Psychology, Motol University Hospital, Prague, Czech Republic
* These authors contributed equally to this work.

Corresponding author
Pavel Krsek, pavel.krsek@fnmotol.cz

## ABSTRACT

**Background.** We aimed first to describe trends in cognitive performance over time in a large patient cohort ($n = 203$) from a single tertiary centre for paediatric epilepsy surgery over the period of 16 years divided in two (developing—pre-2011 vs. established—post-2011). Secondly, we tried to identify subgroups of epilepsy surgery candidates with distinctive epilepsy-related characteristics that associate with their pre- and post-surgical cognitive performance. Thirdly, we analysed variables affecting pre-surgical and post-surgical IQ/DQ and their change (post- vs. pre-surgical).

**Methods.** We analysed IQ/DQ data obtained using standardized neuropsychological tests before epilepsy surgery and one year post-surgically, along with details of patient's epilepsy, epilepsy surgery and outcomes in terms of freedom from seizures. Using regression analysis, we described the trend in post-operative IQ/DQ. Cognitive outcomes and the associated epilepsy- and epilepsy surgery-related variables were compared between periods before and after 2011. Using multivariate analysis we analysed the effect of individual variables on pre- and post-operative IQ/DQ and its change.

**Results.** Epilepsy surgery tends to improve post-surgical IQ/DQ, most significantly in patients with lower pre-surgical IQ/DQ, and post-surgical IQ/DQ strongly correlates with pre-surgical IQ/DQ (Rho $= 0.888$, $p < 0.001$). We found no significant difference in pre-, post-surgical IQ/DQ and IQ/DQ change between the periods of pre-2011 and post-2011 ($p = 0.7$, $p = 0.469$, $p = 0.796$, respectively). Patients with temporal or extratemporal epilepsy differed in their pre-surgical IQ/DQ ($p = 0.001$) and in IQ/DQ change ($p = 0.002$) from those with hemispheric epilepsy, with no significant difference in post-surgical IQ/DQ ($p = 0.888$). Groups of patients with different underlying histopathology showed significantly different pre- and post-surgical IQ/DQ ($p < 0.001$

and $p < 0.001$ respectively) but not IQ/DQ change ($p = 0.345$). Variables associated with severe epilepsy showed effect on cognitive performance in multivariate model. **Discussion**. Post-surgical IQ/DQ strongly correlates with pre-surgical IQ/DQ and greatest IQ/DQ gain occurs in patients with lower pre-surgical IQ/DQ scores. Cognitive performance was not affected by changes in paediatric epilepsy surgery practice. Pre- and post-operative cognitive performances, as well as patients' potential for cognitive recovery, are highly dependent on the underlying aetiology and epileptic syndrome.

## INTRODUCTION

Epilepsy surgery represents an established method for treatment of focal drug resistant epilepsy in both children and adults (*Ryvlin, Cross & Rheims, 2014*). The major aims of paediatric epilepsy surgery are: to relieve the patient of debilitating seizures and anti-epileptic medication, or at least to significantly decrease seizure frequency, and to prevent further decline in developmental/cognitive functions, often accompanying drug resistant epilepsy (*Freitag & Tuxhorn, 2005*; *Moosa & Wyllie, 2017*). In the past few decades, the outcome in terms of freedom from seizures has either tended to improve (*Hemb et al., 2010*), or remained stable (*Lamberink et al., 2015*) in large paediatric epilepsy surgery centres. Freedom from seizures has repeatedly been associated with favourable cognitive outcome (*Puka, Tavares & Smith, 2017*; *Van Schooneveld & Braun, 2013*; *Viggedal et al., 2013*), even with catch-up in mental development (*Freitag & Tuxhorn, 2005*).

In an extensive review (*Van Schooneveld & Braun, 2013*), multiple pre-, post-surgical and surgery-related variables affecting cognitive performance before and after epilepsy surgery were identified, including epilepsy duration, underlying aetiology, age at surgery, seizure outcome, etc. Multiple additional factors play their role in cognitive development of children with focal structural epilepsy, such as the extent of the epileptogenic zone and of the zone of dysfunction, the epileptiform discharges and the effects of antiepileptic medication. Their interacting effects may influence patients' long-term prognosis to achieve freedom from seizures and optimal cognitive development (*Moosa & Wyllie, 2017*).

Longitudinal studies on seizure outcomes of children undergoing epilepsy surgery compared seizure outcomes between early vs. late period of the epilepsy surgery program (*Hemb et al., 2010*; *Lamberink et al., 2015*). The results reflected changes in epilepsy surgery practice: the advent of novel diagnostic methods (magnetoencephalography, post-processing neuroimaging methods, source imagingand others) and improved surgical techniques (e.g., increased use of stereo-EEG). However, none of these studies focused on differences in cognitive outcomes between the two periods.

Most studies on cognitive outcomes of paediatric epilepsy surgery patients have been limited to small patients series even though over a long follow-up period (*Viggedal et al., 2012*) or to specific subgroups, e.g., patients with low IQ (*Malmgren et al., 2008*),

preschool children (*Freitag & Tuxhorn, 2005*), children undergoing hemispherotomy for hemimegalencephaly (*Honda et al., 2013*). *Skirrow et al. (2011)* analysed cognitive outcomes in children with temporal lobe epilepsy and provided evidence that resective epilepsy surgery leads to long-term cognitive improvement, compared to medical treatment only. *Lee et al. (2014)* showed that even patients with severe epileptic encephalopathy, including Lennox-Gastaut and West syndrome, may profit from epilepsy surgery both in terms of seizure and cognitive outcome; in 73.5% of cases malformations of cortical development (MCD) represented the underlying aetiology. *Honda et al. (2013)* have proven the positive effect of early hemispherotomy for hemimegalencephaly on developmental outcome of post-surgically seizure-free children.

A more recent longitudinal study (*Sibilia et al., 2017*) followed 31 children who underwent epilepsy surgery and a control group of 14 medically treated paediatric epilepsy surgery candidates and found no significant difference between the groups in IQ/DQ both at the beginning of the study and after two years of follow-up. The developmental trajectories, however, differed, and patients in the surgical group improved, while patients in the conservative group showed decrease in IQ/DQ over the two-year period. The main limitation of this study, however, remains its small sample size. The problems of sample size and the length of follow-up period were addressed in another longitudinal study (*Puka, Tavares & Smith, 2017*) that analysed cognitive performance in a surgical and a non-surgical group of 97 paediatric epilepsy patients and unexpectedly found no difference in cognitive performance between the groups after a follow-up period of 4 to 11 years. The control group, however, was quite heterogeneous, combining potential surgical candidates who refused surgery and those who were not indicated for epilepsy surgery in the first place, with varying aetiologies.

To summarize, many studies on cognitive outcomes in paediatric epilepsy surgery patients are limited by small sample size, short follow-up period, and they often fail to distinguish patients' cognitive abilities according to the underlying aetiology.

In this study we aimed first to analyse trends in cognitive profiles of children undergoing epilepsy surgery over the period of 17 years and to test whether there were differences in cognitive performance (pre-surgical IQ, post-surgical IQ, or change from pre-surgical to post-surgical IQ) between the early and late period of the paediatric epilepsy surgery program in our tertiary clinic. Next, we assessed whether there were differences in cognitive performance between the early and the late period for specific underlying aetiologies and epilepsy syndromes. Finally, we aimed to identify variables affecting cognitive performance in the entire cohort.

## MATERIALS & METHODS

### Patient selection

Paediatric patients ($\leq$19 years of age) investigated for and having undergone epilepsy surgery in Motol Epilepsy Centre between January 1st, 2000 and December 31st, 2017 with available data on (i) results of pre- and post-surgical neuropsychological evaluation, (ii) seizure outcome one year after epilepsy surgery in patients included in analysis of post-surgical IQ/DQ and on (iii) pre- and post-surgical epilepsy- and epilepsy surgery-related

variables were included in the study. Patients who underwent multiple epilepsy surgeries were excluded. The dataset was completed to our best knowledge; however, due to the nature and length of the study, some data might have been missed. Since the study is observational in nature and no experimental procedures were performed, the approval of Motol University Hospital ethics committee was not required. Informed consent with the pre-surgical evaluation and epilepsy surgery was obtained prior to all procedures from patients or their legal representatives.

## Study design

We retrospectively analysed pre-, post-surgical IQ/DQ and the change between post- and pre-surgical IQ/DQ in relation to multiple epilepsy- and epilepsy surgery-related variables. In addition, the study period was divided in two—pre-2011 and post-2011—to compare trends in cognitive outcomes between the periods of developing vs. established epilepsy surgery program, similar to methodology published elsewhere (*Lamberink et al., 2015*). The cut-off date was set to December 31st, 2010. The year 2011 was chosen as the dividing line for several reasons: it was marked by the introduction of stereo-EEG as a method of long-term invasive EEG monitoring in our centre, and the number of epilepsy surgery procedures stabilized after the previously observed increase every year (*Belohlavkova et al., 2019*).

By comparing two periods of epilepsy surgery program, we aimed to analyse whether changes in patient population (e.g., different distribution of aetiologies, epileptic syndrome, age at surgery, etc.) and novel diagnostic and treatment strategies (e.g., use of stereo-EEG and advances in neuroimaging) that we report elsewhere (*Belohlavkova et al., 2019*) were reflected in cognitive performance of patients included in the study. All studied epilepsy-related variables are listed in Supplemental Information. Solely for the purpose of this study we used the term "epileptic syndrome" to denote either temporal, or extratemporal or hemispheric electro-clinical epileptic syndrome without any relationship to the underlying structural or genetic aetiology; hypothalamic hamartomas were excluded for the purpose of analysis of epileptic syndromes. "Abnormal neurological finding" stands for any focal abnormality in standard neurological examination. Surgery type represents the type of epilepsy surgery procedure: hemispherectomy/hemispherotomy, tailored resection, lesionectomy, extended lesionectomy, standardized resection (in our series exclusively anterior mesial temporal resection); surgery extent denotes the extent of resection: hemispheric, focal, lobar or multilobar. Surgery location may be either left-, right-sided or midline (hypothalamic hamartomas were preserved for this analysis). Complications related to epilepsy surgery were classified as major or minor as published elsewhere (*Bjellvi et al., 2015*). Types of focal cortical dysplasia (FCD) were classified according to ILAE consensus classification (*Blumcke et al., 2011*). FCD type Ia, Ib and Ic are characterized by abnormalities in radial, tangential and radial and tangential migration. FCD type IIa and IIb display cytological abnormalities, including dysmorphic neurons (FCD IIa and IIb) and balloon cells (FCD IIb). FCD type III occurs adjacent to another pathology: hippocampal sclerosis (FCD IIIa), glial or glioneural tumour (FCD IIIb), vascular malformation (FCD IIIc) or lesion acquired early in life (FCD IIId)

(*Blumcke et al., 2011*). In one type of analysis, histopathological lesions were divided in two groups: developmental vs. acquired. MCD, TSC and long-term epilepsy-associated tumours (LEAT) were considered developmental, and inflammatory, post-traumatic and glial scar lesions were considered acquired. Specifically for the purpose of this analysis, hippocampal sclerosis was excluded, as the proportion of its developmental or acquired origin has not yet been unequivocally established.

Neuropsychological examination was performed before the (first) surgery and at one year follow-up after the resection using (according to patient's age and cognitive level) Wechsler Adult Intelligence Scale in 3rd revision (*Wechsler, 2010*), Wechsler Intelligence Scale for Children in 3rd revision (*Wechsler, 2002*) for evaluating IQ and Bayley Scales of Infant Development in 2nd revision (*Bayley, 1993*) for developmental quotient assessment. Stanford-Binet Intelligence Scale in 4th revision (*Thorndike, Hagen & Sattler, 1986*) was selected for IQ testing in children with lower cognitive skills (expected IQ < 60), despite its limited reproducibility and reliability of IQ testing in this group of patients.

## Statistical procedures

Pre-surgical and post-surgical IQ/DQ and IQ/DQ change scores were selected as dependent variables. We first performed regression analysis to describe the relationship between pre- and post-surgical IQ/DQ in the entire cohort and separately for children above and below 6 years of age and for children above and below 12 years of age. Then, we did univariate regression analysis for continuous observed (independent) variables and selected those with $p$-value below 0.05 for multivariate analysis. We performed one-way ANOVA for categorical variables and calculated median difference and its 95% confidence interval using Hodges-Lehman estimator. For calculation of effect sizes, omega-squared effect size was used for ANOVA tests and r-squared effect size for correlation tests and multivariant models.

In the multivariate analysis, we calculated a multiple regression based on the general linear model by stepwise regression algorithm with the variables that reached statistical significance in univariate testing. The beta coefficients with $p$-values of < 0.05 were considered statistically significant. For the calculations software MatLab version 2017b and its statistical computing toolbox was used.

## RESULTS

A total of 203 patients were included in the study (103 males, 100 females). For patients' details see Supplementary materials and Table 1. After accounting for the missing data, 191 patients were included in the analyses of pre-surgical IQ/DQ, 156 patients of post-surgical IQ/DQ and 154 of IQ/DQ change.

### Trends in cognitive performance in the periods of pre-2011 vs. post-2011

Using regression analysis, we have shown there exists strong and significant correlation between pre- and post-surgical IQ/DQ, and that patients undergoing epilepsy surgery tend to have higher post-surgical IQ/DQ than pre-surgical IQ/DQ (Rho = 0.888, $p < 0.001$).

**Table 1** **Demographic features of patients included in the study.** The table shows demographic features of patients included in the study.

| | Pre-2011 [count] | Pre-2011 [%] | Post-2011 [count] | Post-2011 [%] | Overall [count] | Overall [%] |
|---|---|---|---|---|---|---|
| **Sex** | | | | | | |
| Female | 40 | 51.3 | 60 | 48.0 | 100 | 49.3 |
| Male | 38 | 48.7 | 65 | 52.0 | 103 | 50.7 |
| **Family history of epilepsy** | 10 | 12.8 | 20 | 16.0 | 30 | 14.8 |
| **Prenatal risks** | 11 | 14.1 | 16 | 12.8 | 27 | 13.3 |
| **Febrile seizures** | 12 | 15.4 | 5 | 4.0 | 17 | 8.4 |
| **Trauma** | 4 | 5.1 | 1 | 0.8 | 5 | 2.5 |
| **Inflammation** | 3 | 3.9 | 3 | 2.4 | 6 | 3.0 |
| **Infantile spasms** | 5 | 6.4 | 15 | 12.0 | 20 | 9.9 |
| **Epileptic syndrome** | | | | | | |
| Temporal lobe epilepsy | 39 | 50.0 | 54 | 43.2 | 93 | 45.8 |
| Extratemporal epilepsy | 37 | 47.4 | 57 | 45.6 | 94 | 46.3 |
| Hemispheric epilepsy | 2 | 2.6 | 14 | 11.2 | 16 | 7.9 |
| **Seizure frequency** | | | | | | |
| Daily | 47 | 60.3 | 83 | 66.4 | 130 | 64.0 |
| Less than monthly | 19 | 24.4 | 10 | 8.0 | 29 | 14.3 |
| Monthly | 7 | 9.0 | 8 | 6.4 | 15 | 7.4 |
| Weekly | 5 | 6.4 | 24 | 19.2 | 29 | 14.3 |
| **Status epilepticus** | 11 | 14.1 | 14 | 11.2 | 25 | 12.3 |
| **Focal to bilateral tonic clonic seizures** | 39 | 50.0 | 47 | 37.6 | 86 | 42.4 |
| **Abnormal neurological finding** | 28 | 35.9 | 33 | 26.4 | 61 | 30.1 |
| **MRI finding** | | | | | | |
| Lesional | 73 | 93.6 | 122 | 97.6 | 195 | 96.1 |
| Non-lesional | 5 | 6.4 | 3 | 2.4 | 8 | 3.9 |
| **Surgery type** | | | | | | |
| Extended lesionectomy | 16 | 20.5 | 20 | 16.0 | 36 | 17.7 |
| Hemispherectomy/hemispherotomy | 2 | 2.6 | 15 | 12.0 | 17 | 8.4 |
| Tailored resection | 32 | 41.0 | 47 | 37.6 | 79 | 38.9 |
| Lesionectomy | 7 | 9.0 | 14 | 11.2 | 21 | 10.3 |
| Standardized resection | 21 | 26.9 | 29 | 23.2 | 50 | 24.6 |
| **Surgery extent** | | | | | | |
| Focal resection | 24 | 30.8 | 57 | 45.6 | 81 | 39.9 |
| Hemispherectomy/hemispherotomy | 2 | 2.6 | 11 | 8.8 | 13 | 6.4 |
| Multilobar resection | 12 | 15.4 | 13 | 10.4 | 25 | 12.3 |
| Unilobar resection | 40 | 51.3 | 44 | 35.2 | 84 | 41.4 |
| **Surgery localization** | | | | | | |
| Left hemisphere | 39 | 50.0 | 66 | 52.8 | 105 | 51.7 |
| Midline | 1 | 1.3 | 6 | 4.8 | 7 | 3.5 |
| Rigt hemisphere | 38 | 48.7 | 53 | 42.4 | 91 | 44.8 |
| **Complete resection** | 59 | 75.6 | 105 | 84.0 | 164 | 80.8 |

**Table 1** (*continued*)

| | Pre-2011 [count] | Pre-2011 [%] | Post-2011 [count] | Post-2011 [%] | Overall [count] | Overall [%] |
|---|---|---|---|---|---|---|
| **Histopathology** | | | | | | |
| Encephalitis | 1 | 1.3 | 3 | 2.4 | 4 | 2.0 |
| Hippocampal sclerosis | 20 | 26.0 | 12 | 9.8 | 32 | 16.0 |
| Glial scar | 6 | 7.8 | 6 | 4.9 | 12 | 6.0 |
| MCD | 24 | 31.2 | 36 | 29.3 | 60 | 30.0 |
| Tuberous sclerosis complex | 6 | 7.8 | 12 | 9.8 | 18 | 9.0 |
| Vascular lesion | 0 | 0.0 | 3 | 2.4 | 3 | 1.5 |
| Hypothalamic hamartoma | 1 | 1.3 | 6 | 4.9 | 7 | 3.5 |
| Normal | 0 | 0.0 | 4 | 3.3 | 4 | 2.0 |
| Tumour | 19 | 24.7 | 41 | 33.3 | 60 | 30.0 |
| Not available | | | | | 3 | |
| **FCD class** | | | | | | |
| FCD1 | 7 | 14.6 | 8 | 11.9 | 15 | 13.0 |
| FCD2A | 0 | 0.0 | 6 | 9.0 | 6 | 5.2 |
| FCD2B | 15 | 31.3 | 14 | 20.9 | 29 | 25.2 |
| FCD3A | 11 | 22.9 | 10 | 14.9 | 21 | 18.3 |
| FCD3B | 13 | 27.1 | 21 | 31.3 | 34 | 29.6 |
| Unspecified_MCD | 2 | 4.2 | 8 | 11.9 | 10 | 8.7 |
| **Early post-operative seizures** | 8 | 10.4 | 17 | 13.6 | 25 | 12.4 |
| **Complication type** | | | | | | |
| Major | 4 | 40.0 | 6 | 37.5 | 10 | 38.5 |
| Minor | 6 | 60.0 | 10 | 62.5 | 16 | 61.5 |
| **FCD simplified classification** | | | | | | |
| FCD1 | 7 | 15.2 | 8 | 13.6 | 15 | 14.3 |
| FCD2 | 15 | 32.6 | 20 | 33.9 | 35 | 33.3 |
| FCD3 | 24 | 52.2 | 31 | 52.5 | 55 | 52.4 |
| **Epoch** | | | | | | |
| Before 2011 | 78 | 100.0 | 0 | 0.0 | 78 | 38.4 |
| After 2011 | 0 | 0.0 | 125 | 100.0 | 125 | 61.6 |
| **1 year follow-up period** | | | | | | |
| Seizure-free | 66 | 86.8 | 106 | 89.8 | 172 | 88.7 |
| Seizure reduction < 50% | 6 | 7.9 | 6 | 5.1 | 12 | 6.2 |
| Seizure reduction ≥ 50% | 0 | 0.0 | 2 | 1.7 | 2 | 1.0 |
| Seizure reduction ≥ 90% | 4 | 5.3 | 4 | 3.4 | 8 | 4.1 |
| Not available | | | | | 9 | |

**Notes.**

TLE, temporal lobe epilepsy; XTLE, extratemporal lobe epilepsy; HEMI, hemispheric epilepsy; FCD, focal cortical dysplasia; MCD, malformations of cortical development; TSC, tuberous sclerosis complex; FBTCS, focal to bilateral tonic clonic seizure; LEAT, long-term epilepsy-associated tumours.
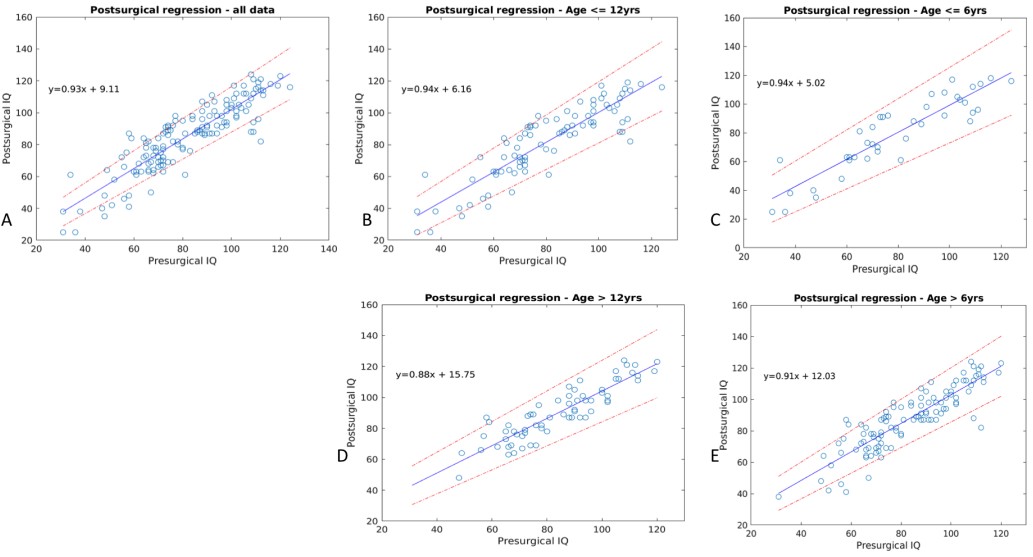

**Figure 1 Result of the regression analysis.** The figure depicts correlation of pre- and post-surgical IQ/DQ scores, along with the respective confidence intervals (dashed line) in the whole dataset, in children above and below 6 and 12 years of age. The respective regression equations are also listed. Specific patients' subgroups are listed in A–E.

The gain in IQ/DQ is more prominent in those with lower pre-surgical IQ/DQ and the group of children older than 6 years of age (Fig. 1). We found no significant difference in pre-, post-surgical IQ/DQ and IQ/DQ change between the periods of pre-2011 and post-2011 ($p = 0.62$, $p = 0.65$, $p = 0.77$, respectively). When analysed in groups according to the epileptic syndrome, differences in cognitive performance (pre-, post-surgical IQ/DQ and IQ/DQ change) between the two periods (pre-2011 vs. post-2011) were not significant for groups of patients with temporal lobe epilepsy (TLE), extratemporal (XTLE) and with hemispheric (HEMI) epilepsy. The same is true in relation to underlying histopathology and developmental vs. acquired lesions (see Supplementary materials). In the analysis of differences between the two periods, we classified FCD in three classes (FCD type I, FCD type II, FCD type III) as the small numbers in respective subcategories (FCD type Ia-Ic, IIa and IIb and IIIa–IIId) would preclude statistical calculation. We found a significantly lower pre- and post-surgical IQ/DQ in the period post-2011 ($F = 8.28$, $p = 0.013$ and $F = 4.76$, $p = 0.05$, respectively) in the group of FCD type I. We did not identify significant difference in post-surgical IQ/DQ and IQ/DQ change between the two periods in either seizure-free or non-seizure-free patient cohort (seizure free: $F = 0.48$, $p = 0.49$ and $F = 0.24$, $p = 0.62$, respectively, non-seizure-free: $F = 0.8$, $p = 0.38$ and $F = 0.26$, $p = 0.62$). For statistical details, including measures of effect size and degrees of freedom, see Supplementary materials.

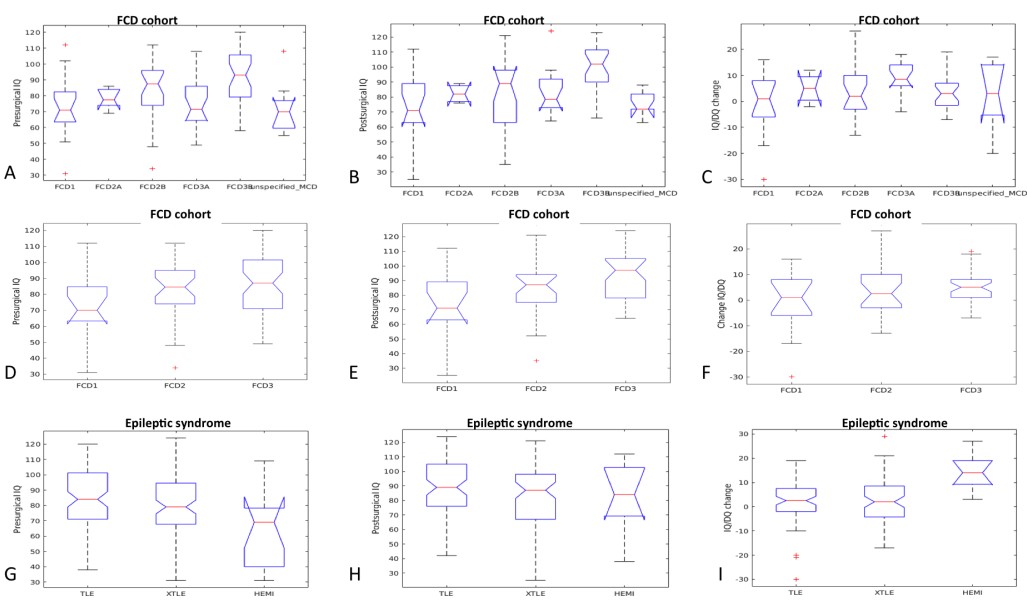

**Figure 2** **Pre-and post-surgical IQ-DQ and IQ/DQ change in distinct groups of patients according to epilepsy-related features.** Specific patients' subgroups are listed in A–I. FCD, focal cortical dysplasia; MCD, malformations of cortical development; TSC, tuberous sclerosis complex; TLE, temporal lobe epilepsy; XTLE, extratemporal lobe epilepsy; HEMI, hemispheric epilepsy.

## Pre- and post- surgical cognitive performance in distinct groups of patients according to their epilepsy-related features

Using ANOVA statistics we observed that patients with TLE differ from those with XTLE and those with HEMI epilepsy in their pre-surgical IQ/DQ ($F = 6.83$, $p = 0.001$) and in their IQ/DQ change ($F = 6.63$, $p = 0.002$); post-hoc tests showed that patients with HEMI score significantly lower in their IQ/DQ compared to those with XTLE and TLE (Fig. 2). Depending on their underlying histopathology, the analyses have identified group differences in pre- and post-surgical IQ/DQ ($F = 6.56$, $p < 0.001$ and $F = 7.44$, $p < 0.001$, respectively). Post-hoc analyses found significantly higher pre-surgical IQ/DQ in patients with tumour (LEAT) compared to those with MCD ($p = 0.002$), TSC ($p < 0,001$) and glial scar ($p = 0.005$). Significant differences in post-surgical IQ/DQ were also observed between the aetiological sub-groups; patients with LEAT achieve significantly higher IQ/DQ scores than those with e.g., MCD, TSC, and those with hippocampal sclerosis score higher than patients with TSC. When comparing developmental vs. acquired histopathological lesions we found no significant difference in their pre-, post-surgical IQ/DQ and IQ/DQ change (data not shown). Furthermore, we analysed the differences in cognitive performance relative to FCD classes (*Blumcke et al., 2011*), and found significant difference between the classes in the pre- and post-surgical IQ/DQ ($F = 4.89$, $p < 0.001$ and $F = 4.73$, $p < 0.001$, respectively); the results remained significant for post-surgical IQ/DQ for the classes of FCD type I, II and III ($F = 6.32$, $p = 0.003$). We observed a trend towards higher pre-surgical IQ/DQ and IQ/DQ change in FCD type III vs. FCD type II and type I (Fig. 1).

**Table 2  Table showing results of univariate testing of categorical variables.** The table shows categorical variables associated with pre- (A) and post-surgical IQ/DQ (B) and IQ/DQ change (C).

| Variable name | Median difference (95% CI) | Mean difference (95% CI) | p-value | Effect size | Fstat | df |
|---|---|---|---|---|---|---|
| **A: Determinants of pre-surgical IQ/DQ** ($n = 191$) | | | | | | |
| Prenatal risks | 8 (0,17) | 9.0(1,17) | 0,031 | 0,019 | 4,722 | 189 |
| Infantile spasms | 20 (9,30) | 19.1(10,28) | <0.001 | 0,070 | 15,305 | 189 |
| Epileptic syndrome | | | 0,001 | 0,058 | 6,832 | 188 |
| Seizure frequency | | | 0,035 | 0,029 | 2,921 | 187 |
| Status epilepticus | 11 (2,20) | 10.7(2,19) | 0,018 | 0,024 | 5,725 | 189 |
| Abnormal neurological finding | 18 (13,25) | 18.5(13,24) | <0.001 | 0,171 | 40,231 | 189 |
| Histopathology | | | <0.001 | 0,184 | 6,281 | 189 |
| FCD class | | | 0,001 | 0,145 | 4,573 | 189 |
| **B: Determinants of post-surgical IQ/DQ** ($n = 156$) | | | | | | |
| Infantile spasms | 26 (13,39) | 25.6(15,36) | <0.001 | 0,121 | 22,383 | 154 |
| Seizure frequency | | | 0,014 | 0,048 | 3,631 | 152 |
| Status epilepticus | 15 (5,25) | 14.8(4,26) | 0,010 | 0,036 | 6,855 | 154 |
| Abnormal neurological finding | 17 (9,25) | 16.6(10,23) | <0.001 | 0,119 | 21,928 | 154 |
| Surgery type | | | <0.001 | 0,109 | 5,724 | 151 |
| Surgery extent | | | <0.001 | 0,118 | 7,902 | 152 |
| Complete resection | −14(−21,−6) | −13.3(−21,−6) | 0,001 | 0,062 | 11,268 | 154 |
| Histopathology | | | <0.001 | 0,243 | 7,129 | 145 |
| FCD class | | | 0,001 | 0,177 | 4,572 | 78 |
| FCD1 | | | 0,008 | 0,072 | 7,443 | 82 |
| FCD3B | | | <0.001 | 0,178 | 19,009 | 82 |
| 1 year follow-up period | | | 0,035 | 0,037 | 2,953 | 150 |
| Age class12 | −6(−12,0) | −7.5(−14,−1) | 0,027 | 0,025 | 4,956 | 154 |
| Age class6 | −7(−16,0) | −8.7(−16,−2) | 0,024 | 0,026 | 5,167 | 154 |
| **C: Determinants of IQ/DQ change** ($n = 154$) | | | | | | |
| Epileptic syndrome | | | 0,002 | 0,069 | 6,635 | 151 |
| Surgery type | | | 0,025 | 0,046 | 2,863 | 149 |
| Surgery extent | | | 0,011 | 0,052 | 3,822 | 150 |
| FCD1 | | | 0,047 | 0,037 | 4,080 | 80 |
| FCD3A | | | 0,022 | 0,052 | 5,430 | 80 |
| Age class12 | −4(−7,−2) | −4.6(−7,−2) | 0,004 | 0,048 | 8,795 | 152 |
| Age class6 | −4(−8,−1) | −4.0(−7,-1) | 0,027 | 0,025 | 4,961 | 152 |

**Notes.**

CI, confidence interval; SE, standard error; SD, standard deviation; FCD, focal cortical dysplasia; df, degrees of freedom.

## Predictors of pre- and post-surgical IQ/DQ and its change over the entire period

The results of univariate testing are listed in Table 2 for categorical variables and in Table 3 for continuous variables associated with pre-surgical IQ/DQ, post-surgical IQ/DQ and IQ/DQ change; only the variables significant in univariate testing are shown here, all studied variables are listed in Supplementary materials. The used stepwise regression algorithm excludes variables that might skew results in the calculation process of the

**Table 3 Table showing results of univariate testing of continuous variables.** The table shows continuous variables associated with pre- (A) and post-surgical IQ/DQ (B) and IQ/DQ change (C).

**A: Determinants of presurgical IQ/DQ (*n* = 191)**

| Variable | Age of first seizure |
|---|---|
| | Univariate regression |
| **Intercept** | 73.9 |
| SE | 2 |
| *t* statistics | 36.88 |
| *p* value | <0.001 |
| **beta 1** | 1.55 |
| SE | 0.3 |
| *t* statistics | 5.2 |
| *p* value | <0.001 |
| dfe | 189 |
| sse | 65805 |
| dfr | 1 |
| ssr | 9416 |
| f | 27.05 |
| *p* value | <0.001 |
| r2 effect size | 0.13 |

**B: Determinants of postsurgical IQ/DQ (*n* = 156)**

| Variable | Age of first seizure | Age at surgery | Duration of epilepsy |
|---|---|---|---|
| | Univariate regression | | |
| **intercept** | 75.6 | 77.87 | 90 |
| SE | 2.3 | 3.56 | 2.65 |
| *t* statistics | 33.05 | 21.86 | 33.9 |
| *p* value | <0.001 | <0.001 | <0.001 |
| **beta 1** | 1.99 | 0.7 | 0.8 |
| SE | 0.35 | 0.3 | 0.36 |
| *t* statistics | 5.78 | 2.38 | −2.3 |
| *p* value | <0.001 | 0.02 | 0.02 |
| dfe | 154 | 154 | 154 |
| sse | 57589 | 67579 | 67798 |
| dfr | 1 | 1 | 1 |
| ssr | 12474 | 2484 | 2264 |
| f | 33.36 | 5.66 | 5.14 |
| *p* value | <0.001 | 0.02 | 0.02 |
| r2 effect size | 0.18 | 0.04 | 0.03 |

**C: Determinants of IQ/DQ change (*n* = 154)**

| Variable | Age at surgery | Duration of epilepsy |
|---|---|---|
| | Univariate regression | |
| **Intercept** | −0.4 | 2.19 |

*(continued on next page)*

**Table 3** (*continued*)

**C: Determinants of IQ/DQ change (*n* = 154)**

| Variable | Univariate regression | |
| --- | --- | --- |
| | **Age at surgery** | **Duration of epilepsy** |
| SE | 1.66 | 1.25 |
| *t* statistics | −0.24 | 1.75 |
| *p* value | 0.81 | 0.08 |
| **beta 1** | 0.36 | 0.21 |
| SE | 0.14 | 0.17 |
| *t* statistics | 2.58 | 1.23 |
| *p* value | 0.01 | 0.22 |
| dfe | 152 | 152 |
| sse | 14157 | 14629 |
| dfr | 1 | 1 |
| ssr | 617.55 | 145.83 |
| f | 6.63 | 2 |
| *p* value | 0.01 | 0.22 |
| r2 effect size | 0.04 | 0.01 |

**Notes.**

SE, standard error; dfe, degrees of freedom for error; sse, sum of squares for error; dfr, degrees of freedom for regression; ssr, sum of squares for regression.

multiple regression based on the general linear model; therefore, some variables, despite being significant in univariate testing, do not appear in the results of multivariate testing, listed in Table 4. Some variables associated with severe epilepsy, e.g., the presence of infantile spasms affecting pre-surgical IQ/DQ remained significant in univariate testing only. The factors found significant in multiple regression model were (i) the presence of abnormal neurological finding and FCD type IIb and IIIb and younger age at first seizure affecting pre-surgical IQ/DQ; (ii) age at first seizure, status epilepticus, multilobar resection, unilobar resection, complete resection and TSC affecting post-surgical IQ/DQ; (iii) age at surgery and temporal or extratemporal epileptic syndrome affecting IQ/DQ change (see Table 4). The date of surgery, reflecting change in epilepsy surgery practice, was not rendered significant in the multivariate regression analysis model and was therefore excluded from the model by the used stepwise algorithm.

For the summary of statistically significant results of all tests, including post-hoc analyses and the respective *p* values, F values, effect sizes and degrees of freedom, see Tables 1–5.

## DISCUSSION

In the period of 2000–2017 we collected data on 203 children who underwent resective epilepsy surgery in the paediatric part of Motol Epilepsy Centre and studied their cognitive performance in relation to the epilepsy- and epilepsy surgery-related characteristics and seizure outcome.

We have observed a strong correlation between the pre- and post-surgical IQ/DQ, in accordance with multiple previous studies that have shown pre-surgical IQ/DQ to be a strong independent predictor of post-surgical IQ/DQ (*Puka, Tavares & Smith, 2017*;

**Table 4** **Table showing results of multivariate testing.** The table shows the results and features of multiple regression based on the general linear model of pre-(A), post-surgical IQ/DQ (B) and IQ/DQ change (C).

| Variable | Estimate | SE | t statistics | p value |
|---|---|---|---|---|
| **A: Multiple regression based on the general linear model of pre-surgical IQ/DQ** | | | | |
| Intercept | 71.61 | 4.84 | 14.81 | <0.001 |
| Age at first seizure | 0.82 | 0.38 | 2.18 | 0.03 |
| Abnormal neurological finding | −20.08 | 5.03 | −3.99 | <0.001 |
| FCD IIa | 6.95 | 7.33 | 0.95 | 0.35 |
| FCD IIb | 14.29 | 4.97 | 2.87 | 5 |
| FCD IIIa | 1.02 | 5.21 | 0.2 | 0.84 |
| FCD IIIb | 14.72 | 4.79 | 3.07 | 3 |
| Unspecified MCD | 12.75 | 6.82 | 1.87 | 0.06 |
| **Model characteristics** | **Value** | | | |
| Observations | 106 | | | |
| Error degrees of freedom | 97 | | | |
| Estimated dispersion | 221 | | | |
| F-statistic vs. constant model | 7.77 | | | |
| p value | <0.001 | | | |
| r2 effect size | 0.32 | | | |
| **B: Multiple regression based on the general linear model of post-surgical IQ/DQ** | | | | |
| Intercept | 93.6 | 11.55 | 8.1 | <0.001 |
| Age at first seizure | 0.82 | 0.34 | 2.42 | 0.02 |
| Status epilepticus | −11.27 | 4.76 | −2.37 | 0.02 |
| Hemispherectomy/hemispherotomy | −12.59 | 7.74 | −1.63 | 0.11 |
| Multilobar resection | −18.41 | 4.63 | −3.98 | <0.001 |
| Unilobar resection | −9.06 | 3.51 | −2.58 | 0.01 |
| Complete resection | 9.34 | 3.76 | 2.48 | 0.01 |
| Hippocampal sclerosis | −9.61 | 11.35 | −0.85 | 0.4 |
| Glial scar | −10.65 | 11.64 | −0.92 | 0.36 |
| MCD | −16.69 | 10.68 | −1.56 | 0.12 |
| TSC | −30.83 | 11.38 | −2.71 | 0.01 |
| Vascular lesion | 4.03 | 15.83 | 0.25 | 0.8 |
| Hypothalamic hamartoma | −13.37 | 12.88 | −1.04 | 0.3 |
| Normal histopathology | −14.47 | 14.44 | -1 | 0.32 |
| Tumour | −2.07 | 10.84 | −0.19 | 0.85 |
| **Model characteristics** | **Value** | | | |
| Observations | 154 | | | |
| Error degrees of freedom | 139 | | | |
| Estimated dispersion | 268 | | | |
| F-statistic vs. constant model | 8.54 | | | |
| p value | <0.001 | | | |
| r2 effect size | 0.46 | | | |

**Table 4** (*continued*)

| Variable | Estimate | SE | *t* statistics | *p* value |
|---|---|---|---|---|
| **C: Multiple regression based on the general linear model of IQ/DQ change** | | | | |
| Intercept | 10.91 | 3.28 | 3.33 | 1 |
| TLE | −13.53 | 3.3 | −4.09 | <0.001 |
| XTLE | −12.38 | 3.26 | −3.8 | <0.001 |
| Age at surgery | 0.44 | 0.14 | 3.18 | 2 |
| **Model characteristics** | **Value** | | | |
| Observations | 154 | | | |
| Error degrees of freedom | 150 | | | |
| Estimated dispersion | 84.8 | | | |
| F-statistic vs. constant model | 8.07 | | | |
| *p* value | <0.001 | | | |
| r2 effect size | 0.14 | | | |

Notes.

SE, standard error; TLE, temporal lobe epilepsy; XTLE, extratemporal lobe epilepsy; HEMI, hemispheric epilepsy; FCD, focal cortical dysplasia; MCD, malformations of cortical development; TSC, tuberous sclerosis complex; FBTCS, focal to bilateral tonic clonic seizure; LEAT, long-term epilepsy-associated tumours.

**Table 5** **Summary of post-hoc tests.** (A) Post-hoc tests for variables associated with presurgical IQ/DQ. (B) Post-hoc tests for variables associated with postsurgical IQ/DQ.

| Variable name | Mean difference | Median difference | *p*-value | Effect size | Fstat | df |
|---|---|---|---|---|---|---|
| **A: Post-hoc tests for IQ/DQ_presurgical** | | | | | | |
| Glial scar_tumour | −22.79 (−41.92,−3.67) | −20(−36,−8) | <0.001 | 0.189 | 16.583 | 66 |
| MCD_tumour | −13.84 (−24.72,−2.97) | −14(−20,−7) | <0.001 | 0.130 | 17.613 | 110 |
| TSC_tumour | −29.23 (−45.28, −29.23) | −31(−39,−21) | <0.001 | 0.355 | 41.261 | 72 |
| HS_tumour | −12.86 (−25.82,−12.86) | −14(−22,−5) | 0.001 | 0.109 | 11.683 | 86 |
| FCD1_FCD3B | −19.46 (−35.02,−3.89) | −22(−33,−8) | 0.001 | 0.208 | 12.815 | 44 |
| FCD3A_FCD3B | −17.70 (−32.20,−3.19) | −19(−28,−9) | <0.001 | 0.220 | 15.088 | 49 |
| FCD3B_unspecified MCD | 20.42 (1.28,39.57) | 21(5,34) | 0.006 | 0.168 | 8.664 | 37 |
| TLE_hemi | 20.95 (7.09,34.80) | 21(36,7) | 0.001 | 0.104 | 12.709 | 100 |
| XTLE_hemi | 16.69 (2.84,30.54) | 18(32,3) | 0.005 | 0.066 | 8.175 | 100 |
| **B: Post-hoc tests for IQ/DQ_postsurgical** | | | | | | |
| MCD_tumour | −20.69 (−33.36,−8.02) | −21(−27,−14) | <0.001 | 0.241 | 28.870 | 87 |
| TSC_tumour | −35.71 (−53.29,−18.12) | −35(−47,−24) | <0.001 | 0.457 | 49.766 | 57 |
| HS_TSC | 22.33 (3.10,41.55) | 21(8,35) | 0.001 | 0.221 | 12.379 | 39 |
| Hamartoma_tumour | −23.25 (−45.26,−1.24) | −19(−36,−3) | 0.005 | 0.136 | 8.722 | 48 |
| FCD1_FCD3B | −26.55 (−44.98,−8.13) | −27(−41,−12) | <0.001 | 0.307 | 17.353 | 36 |
| FCD3A_FCD3B | −20.61 (−35.34,−5.89) | −20(−29,−7) | 0.005 | 0.180 | 9.133 | 36 |
| Unspecified MCD_FCD3B | 23.84 (0.31,47.37) | 26(13,36) | 0.002 | 0.275 | 12.005 | 28 |

Notes.

FCD, focal cortical dysplasia; MCD, malformations of cortical development; TSC, tuberous sclerosis complex; HS, hippocampal sclerosis; df, degrees of freedom.

*Van Schooneveld & Braun, 2013*). Our results also confirm previously reported observations that epilepsy surgery leads to greater increase in IQ/DQ in children with lower pre-surgical IQ/DQ (*Puka, Tavares & Smith, 2017*). Loddenkemper *et al.* have shown similar results in infants <3 years of age who presented with lower DQ scores and infantile spasms and

achieved significantly higher DQ scores after surgery. The increase however was especially prominent in children operated on before 1 year of age (*Loddenkemper et al., 2007*). The authors suggest the increase might result from the fact that the deleterious epileptic activity had been terminated early enough to impede development. Therefore, we presume, it is not the value of IQ/DQ *per se* that leads to cognitive improvement after surgery but rather the early indication of epilepsy surgery in infants presenting with severe epilepsy and developmental delay. Cognitive performance of children who enter epilepsy surgery programs with normal or above-average IQ tends neither to improve, nor to worsen with epilepsy surgery. Although the actual numerical increase in IQ/DQ scores did not reach clinically significant values of 8-15 points (*Van Schooneveld & Braun, 2013*), based on the regression curve equation, this might have been mostly due to the limited follow-up period. Our results however, are in line with other studies showing that IQ/DQ scores tend to remain stable or improve after epilepsy surgery in children over a broad spectrum of cognitive performance, and even children with low pre-surgical IQ/DQ do not lose cognitive skills but rather benefit from epilepsy surgery (*Loddenkemper et al., 2007*; *Viggedal et al., 2012*; *Viggedal et al., 2013*). Unexpectedly, we observed greater gain in IQ/DQ scores in children above 6 years of age (Fig. 1), compared to their younger counterparts. Given that these children were operated on later, they also might have had later epilepsy onset, and the effect of drug resistant seizures was not present in early sensitive periods of brain development. Also, the group of children below 6 years of age was rather small which might have skewed the results. In addition, the effect of repeated measurement cannot be completely ruled out (*Sherman et al., 2003*).

## Cognitive performance in children undergoing epilepsy surgery did not change between the periods of pre-2011 and post-2011

Comparing cognitive performance of all epilepsy surgery patients from the period of pre-2011 vs. post-2011 their pre-, post-surgical IQ/DQ and IQ/DQ change scores did not differ significantly. These findings might imply that the spectrum of epilepsy surgery patients has not changed significantly with respect to their cognitive performance. We further analysed whether distinct groups of patients in terms of epileptic syndrome and underlying aetiology differed in their cognitive performance between the two studied periods and again found no significant difference. These results reflect in part the development of paediatric epilepsy surgery practice in the region and in Europe (*Baud et al., 2018*); we are seeing increased referral of more complex patients and a changing population of paediatric epilepsy surgery candidates. While in the early period we encountered either very severe cases (epilepsy surgery as "the last resort") or very straightforward ones (e.g., hippocampal sclerosis) with a very broad span of cognitive performance, in the later period we observed a decline in the straightforward cases (such as hippocampal sclerosis) and an increase in timely referral of complex cases (such as TSC, see Supplementary materials). The shift towards more complex patients, including extratemporal and MRI negative cases, has been observed in centres around Europe (*Baud et al., 2018*), and this might have been reflected in the cognitive performance of epilepsy surgery patients. Naturally, we cannot exclude the effect of small sample size in certain aetiological groups with less frequent pathologies. To summarize,

the aetiological spectrum of epilepsy surgery candidates has changed significantly in terms of the underlying aetiology (see Supplementary materials and *Belohlavkova et al., 2019*) but the overall IQ/DQ scores remained similar.

## Cognitive performance of children undergoing epilepsy surgery varies based on aetiology and epileptic syndrome

Given the broad spectrum of epileptic syndromes and underlying aetiologies in the studied population we hypothesised that these patients enter pre-surgical evaluation as distinct groups with distinct cognitive performances. According to our analyses, patients with hemispheric syndromes perform significantly lower in their IQ/DQ scores than their counterparts with temporal or extratemporal epilepsy; however, they also have the greatest potential for post-surgical increase in IQ/DQ scores, compared to patients with TLE or XTLE. These results are in line with the results from previous studies showing that patients with hemimegalencephaly profit from early epilepsy surgery (*Bulteau, Otsuki & Delalande, 2013*; *Honda et al., 2013*). Considering the aetiological spectrum of epilepsy surgery patients we hypothesised that patients with developmental structural lesions, such as TSC, MCD or long-term epilepsy-associated tumours differ from those with acquired lesions (e.g., inflammation, glial scar, etc.), but our data showed the opposite. It is rather the specific aetiology itself and the extent of involvement (temporal vs. extratemporal vs. hemispheric epilepsy) that distinguish these patients from each other. Patients with benign tumours achieve significantly better pre- and post-surgical IQ/DQ scores than those with other developmental abnormalities, e.g., TSC or MCD. Given their tendency to reach favourable seizure outcome, in addition to good cognitive outcome, they represent a group that might benefit from early referral to epilepsy surgery (*Ramantani et al., 2014*). Patients with TSC, on the other hand, tend to suffer from developmental delay spanning from mild cognitive impairment to profound intellectual disability in 14% to 31% respectively (*Joinson et al., 2003*). Despite on-going discussions about the pathogenesis of cognitive impairment in TSC (*Curatolo et al., 2016*; *Chu-Shore et al., 2010*), most authors agree that adequate seizure control, achieved by antiepileptic medication or epilepsy surgery, in TSC patients leads to improved cognitive outcomes (*Arya et al., 2015*; *Chu-Shore et al., 2010*).

In histopathological FCD subgroups, we observed a tendency towards better cognitive performance in FCD type III, with the most significant difference in relation to FCD type I. Lower IQ/DQ scores and worse outcomes of epilepsy surgery have been repeatedly observed in patients with FCD type I in contrast to FCD type II (*Krsek et al., 2008*; *Krsek et al., 2009*). FCD type III has only recently been distinguished as a separate category of FCD (*Blumcke et al., 2011*); therefore, it is difficult to compare the results of more and less recent studies. Nevertheless, FCD type III from its definition associates with other pathologies, e.g., hippocampal sclerosis (FCD IIIa) or brain tumour (FDC type IIIb), and patients with these pathologies achieved higher IQ/DQ scores which also explains the observed results. We observed lower pre- and post-surgical IQ/DQ in the period post-2011 in the group of FCD type I; this might be a result of the rather small sample size ($n = 15$). Another plausible explanation is that we tend to accept patients with more severe epilepsy and its related adverse cognitive sequelae in the epilepsy surgery program. It remains to be discovered

whether and to what extent molecular genetic changes in patients with FCD influence their prognosis in terms of both seizure and cognitive outcome (*Baldassari et al., 2019*; *Benova & Jacques, 2018*).

## Multiple factors influence cognitive performance in children undergoing epilepsy surgery

In the entire patient cohort we have identified multiple factors that might affect their cognitive performance. Factors associated with severe epilepsy, including early age at onset, frequent seizures, occurrence of infantile spasms and status epilepticus seemed to be associated with lower pre-surgical IQ/DQ scores. Given that patients suffering from infantile spasms tend to improve in their cognitive skills after successful epilepsy surgery (*Asarnow et al., 1997*) low IQ/DQ scores should not preclude timely referral for epilepsy surgery. In fact, patients with low pre-surgical IQ/DQ scores do benefit from epilepsy surgery even though their chance to achieve seizure freedom decreases with lower IQ/DQ scores (*Malmgren et al., 2008*). Overall, our data support the original concept of epileptic encephalopathy wherein epileptic activity itself contributes to cognitive decline (*Berg et al., 2010*). Therefore, in light of our findings supported by multiple studies advocating for epilepsy surgery as a treatment of choice in drug resistant focal epilepsy, we conclude that epilepsy surgery leads to favourable seizure and cognitive outcomes in carefully selected patients (*Dwivedi et al., 2017*; *Harvey et al., 2008*; *Holthausen, Pieper & Kudernatsch, 2013*; *Ryvlin, Cross & Rheims, 2014*; *Van Schooneveld & Braun, 2013*).

To summarize, patients with varying underlying aetiologies tend to differ in their pre- and post-surgical IQ/DQ scores but not in their IQ/DQ change, we therefore suggest that despite the differences in pre-surgical cognitive performance, patients with various underlying aetiologies may all benefit from epilepsy surgery. These findings are also supported by the regression curve equation that shows that patients with lower pre-surgical IQ/DQ tend to profit most from epilepsy surgery. Understandably, they would still achieve significantly lower post-surgical IQ/DQ scores as post-surgical IQ/DQ scores correlate strongly with pre-surgical IQ/DQ.

Admittedly, our study displays certain limitations, the greatest being the short follow-up period of one year. Indeed, some authors show that only longitudinal studies over lengthy study periods provide solid evidence of cognitive improvement after epilepsy surgery, associated with cessation of antiepileptic medication (*Sibilia et al., 2017*; *Skirrow et al., 2011*). On the other hand, the length of follow-up period in previously published studies spans from few months to many years (*Moosa & Wyllie, 2017*; *Van Schooneveld & Braun, 2013*). Most recently published studies do provide evidence of cognitive improvement or improved cognitive trajectory over longer time periods; however, these studies have other limitations, e.g., smaller sample size or they fail to discern between different aetiologies (*Puka, Tavares & Smith, 2017*; *Sibilia et al., 2017*). Most importantly however, in our study, we aimed primarily to compare trends in cognitive performance between the periods of developing vs. established epilepsy surgery centre and to analyse whether epilepsy surgery candidates differ in their cognitive performance based on underlying aetiology and epileptic syndrome. Further longitudinal studies are needed to provide unequivocal evidence that

paediatric epilepsy surgery can reverse the course of epileptic encephalopathy and lead to cognitive recovery or at least prevent further cognitive decline.

Due to repeated neuropsychological testing, we could not have completely eliminated a possible effect of repeated testing. Although some authors claim that the practice effect in paediatric population is not significant (*Westerveld et al., 2000*), others believe it may interfere with the correct evaluation of cognitive outcomes in paediatric epilepsy surgery (*Sherman et al., 2003*). In addition, given the age diversity of our population, our study faced the issue of comparability of various age-adjusted versions of IQ/DQ tests. We believe both of these issued will be addressed in future studies in which we aim to validate our findings over a longer follow-up period with wider intervals between repeated testing on a more homogenous age-groups of children.

Despite rather large sample size, some numbers of patients in individual subgroups remain small and this precluded some further analyses, e.g., the effect of complications of epilepsy surgery on post-surgical IQ/DQ score. However this also shows that overall, major complications of epilepsy surgery occur rarely in our centre as well as others (*Bjellvi et al., 2015*). Our study also lacked data on socioeconomic status of patients' families that might have affected IQ/DQ change; in fact, parents' education seems to independently contribute to IQ/DQ increase after epilepsy surgery (*Meekes et al., 2015*), while income and residence do not have significant effect (*Puka et al., 2016*). Therefore, the effect of families' socioeconomic status on cognitive performance in children undergoing epilepsy surgery warrants further consideration, both in research and in clinical practice.

Another significant limitation concerns the effect of antiepileptic drugs (AED) on post-surgical cognitive outcome. It has been shown that AED withdrawal leads to improved cognitive skills in children undergoing epilepsy surgery, and that IQ scores tend to increase with decreasing number of AED (*Boshuisen et al., 2015b*). Given the short period of post-operative follow-up period we were unable to assess the effect of AED withdrawal as the vast majority of patients remain on pre-surgical dose of AED at least one year after epilepsy surgery. Accumulating evidence advocates for early AED withdrawal (*Boshuisen et al., 2012*; *Braun & Schmidt, 2014*), and a randomized control trial on the effect of early vs. late AED withdrawal on cognitive outcomes is ongoing (*Boshuisen et al., 2015a*).

## CONCLUSION

Epilepsy surgery may lead to improvement of cognitive performance in patients with drug resistant focal epilepsy, especially in those with lower pre-surgical IQ/DQ scores; this warrants early referral to epilepsy surgery centres. Patients with varying aetiologies and epileptic syndromes enter epilepsy surgery evaluation with distinct cognitive performances and diverse potential for recovery and should be counselled accordingly. In future, longitudinal studies with lengthier follow-up periods and inclusion of novel genetic findings may elucidate mechanisms behind cognitive dysfunction in patients with focal drug resistant epilepsy and contribute to more precise prognosis and counselling.

## ACKNOWLEDGEMENTS

The authors would like to thank their respective clinical teams taking care of patients involved and to the patients and their families themselves for their support and collaboration on the study.

### Funding

This work was supported by MH—CZ DRO University Hospital Motol, Prague, Czech Republic (No. 00064203-6005) and by grant CZ.2.16/3.1.00/2402. The funders had no role in study design, data collection and analysis, decision to publish, or preparation of the manuscript.

### Grant Disclosures

The following grant information was disclosed by the authors:
MH—CZ DRO University Hospital Motol, Prague, Czech Republic: 00064203-6005, CZ.2.16/3.1.00/2402.

### Competing Interests

The authors declare there are no competing interests.

### Author Contributions

- Barbora Benova and Anezka Belohlavkova conceived and designed the experiments, performed the experiments, analyzed the data, prepared figures and/or tables, authored or reviewed drafts of the paper.
- Petr Jezdik analyzed the data, prepared figures and/or tables, authored or reviewed drafts of the paper.
- Alena Jahodova performed the experiments.
- Martin Kudr, Vladimir Komarek, Vilem Novak, Petr Liby, Robert Lesko, Michal Tichy, Martin Kyncl and Josef Zamecnik performed the experiments, approved the final draft.
- Pavel Krsek and Alice Maulisova conceived and designed the experiments, performed the experiments, analyzed the data, contributed reagents/materials/analysis tools, authored or reviewed drafts of the paper, approved the final draft.

### Human Ethics

The following information was supplied relating to ethical approvals (i.e., approving body and any reference numbers):

Since no experimental procedures were performed,and all procedures form a part of a standard clinical practice, the Motol University Hospital ethics committee does not require prior authorization to retrospective observational studies, and therefore allows observational studies to be performed without its prior approval.

### Data Availability

The raw data is available as Supplemental Files.

## Supplemental Information

Supplemental information for this article can be found online at http://dx.doi.org/10.7717/peerj.7790#supplemental-information.

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
