# Peer review of "Cognitive performance in distinct groups of children undergoing epilepsy surgery—a single-centre experience"

_PeerJ, doi:10.7717/peerj.7790_

## Round 0.1 · original submission · Major Revisions

Dear Authors, Happy New Year 2019.Please revise the manuscript according to the two peer reviewer comments that are important to maintain the quality and make it citable and reproducible.

·

Basic reporting

1.1 Style, grammar and vocabulary are generally adequate, but there are some places where the manuscript would benefit from language editing (e.g., line 118: ‘In addition to that,…’; line 190: ‘cognitive performance of studied patients’; lines 311-312: ‘with [a] very broad span of cognitive performance, in the later period we observed [a] decline in the straightforward cases (such as hippocampal and [an] increase in timely referral of complex cases’.
1.2 The introduction does provide an overview of the field. However, although this is fairly comprehensive, it is not very to-the-point. It currently reads more like a listing of different results, rather than a brief, topical review of the literature in the field. I would suggest condensing the current approximately one thousand words to something on the order of 500-700 words, or else fully restructuring the introduction to clarify what is the relevance of the information provided and how each of these data points contributes to an overall picture that informs the present study. It is important to note however, that the references provided are representative of the literature in this field.
1.3 References for the assessment instruments used (e.g., WAIS, WISC) should be included.
1.4 The formal structure of the manuscript is correct. However, the structure of the content can (and should) be improved in many places (see 1.2 above).
1.5 The quality of the figures is acceptable – I particularly like the visualization of the confidence intervals in Figure 1. Figure 2 needs clearer labeling but is otherwise ok.
1.6 The tables should be restructured to provide the information in a more condensed form (see also 1.7 and 4.1).
1.7 The proportion information in the tables (second column) should be integrated in a separate table reporting patient demographics and epilepsy variables, i.e., a condensed version of the current Supplementary Table 1. If this creates problems in terms of journal limits on the number of tables then the current Table 4 can be moved to the Supplementary information.
1.8 In line 242 as well as in many tables, p-values and other non-integer numbers are incorrectly reported with commas as decimal separator, rather than periods. At the very least this should be made consistent (because periods are used elsewhere).
1.9 Where results from statistical tests are reported they should always include the essential information that allows reproduction and interpretation. In particular, F-tests require reporting of the degrees of freedom and some measure of effect size, preferably partial eta-squared but in my opinion standard errors or confidence intervals of the effect are acceptable here since they have the advantage of being directly interpretable in terms of the dependent variable.
1.10 The raw data are available at the time of this review (though see 3.5). However, in their declarations the authors appear to suggest that only summary data will be made available.

Experimental design

2.1 The manuscript reports on a retrospective study of pre- and post-surgical IQ in a fairly large single-center cohort of children undergoing resective brain surgery for treatment of intractable epilepsy. The study therefore clearly falls well within the aims and scope of PeerJ.
2.2 The research questions (lines 159-167) are somewhat poorly defined. In particular, the relation between question i) and questions ii)/iii) is unclear. Are these questions simply combined because they happen to be answerable with the present data? Or is there some particular idea behind this combination which I have been unable to identify? In addition, it is not entirely clear to me precisely what is the intended difference between question ii) and question iii), especially in the way they are addressed in the manuscript: the ‘groups’ mentioned in question iii) are effectively just splits based on variables such as those used in question ii).
2.3 Perhaps more importantly, these questions do not fully match the knowledge gaps identified in the previous paragraph (lines 156-158), which refer to aetiology and the identification of adequate controls. Although the question of suitable controls to assess the effect of epilepsy surgery is an important one, it is not addressed by this study. Aetiology is addressed in questions ii) and iii) but only as part of a larger set of variables. To address these issues, I would suggest a slight change by focusing on three interrelated questions: i) are there changes in cognitive outcome over time? Ii) are there changes in patient characteristics over time? Iii) are the results of these analyses related, e.g., is an (hypothetical) increase in patients with very severe forms of epilepsy responsible for an (equally hypothetical) deterioration in cognitive outcome? This requires two additional analyses: a) an analysis of patient characteristics over time, and b) a multiple regression analysis that includes date of surgery as well as variables previously identified as related to cognitive outcome.
2.4 The data collected for this retrospective study are remarkably complete. Nevertheless, as in any study of this kind, there are missing data and this should be addressed, or at the very least acknowledged (cf. the STROBE guidelines mentioned earlier).
2.5 The authors provide evidence of the ethical evaluation and approval of the study they report on, albeit post-hoc. In my personal opinion, there are no ethical objections against a study of this kind, or if there are, they are vastly outweighed by the benefits. In my view, the study also does not conflict with the Declaration of Helsinki in its most recent form, as accepted by the 64th assembly of the World Medical Association (https://www.wma.net/policies-post/wma-declaration-of-helsinki-ethical-principles-for-medical-research-involving-human-subjects/). However, the most literal interpretation of the PeerJ article requirements for human subjects research (https://peerj.com/about/policies-and-procedures/#human-subjects-research) does not allow this kind of study, since this text does not – at least explicitly – allow for any exceptions to the requirement for informed consent, not even for retrospective analysis of anonymized data as done here. Although the author declarations refer to a “patients' consent form used in clinical practice“, I cannot find this among the files.
2.6 For some patients, IQs in the 20s or 30s are reported. The range of modern, standardized intelligence tests typically does not normally extend much below 50. While there are commonly used techniques to calculate IQ scores for children who score below the lower end of an IQ scale, this needs to be clarified in the text, both to ensure reproducibility and also because such techniques do affect (lower) the reliability of the resulting IQ scores.
2.7 It is very difficult (even with the data in hand) to determine exactly how many patients are included in each analysis. To comply with the STROBE guidelines (https://www.strobe-statement.org/index.php?id=strobe-home), this should be made clear for each analysis. For example, whereas the header for Table 1 states that these are results for the whole dataset, the analysis presumably only included the 191 patients for whom pre-surgical IQ was available. Put differently, the sample size of 203 patients reported in line 234 includes some patients who did not meet the inclusion criteria reported in lines 173-176.
2.8 I also find it difficult to determine how many and exactly which analyses were performed. While I feel that correction for multiple comparisons is not appropriate in this case, since control of Type I error would come at the cost of inflated Type II error, it is essential that the reader is able to determine the structure of the analyses and their exact number.

Validity of the findings

3.1 While I can generally confirm the results from the data provided, I am having trouble obtaining identical (or even similar) values for estimates. The meaning of the columns Median_difference, CI_low, and CI_high in Table 1 is also unclear, and the values listed there seem implausible (e.g. 9 values are listed with two zeros as the first two decimals).
3.2 The statistical analysis does not indeed suggest considerable improvements in IQ after surgery as the text suggests (line 299-300). The authors seem to refer to the regression equation corresponding to the top left panel of Figure 1. However, this equation suggests an increase of 9.1 points only for a hypothetical patient with a pre-surgical IQ equal to 0. For more realistic pre-surgical values, the predicted gain would be considerably smaller (and decrease with increasing pre-surgical IQ: pre 50 -> post 0.93 * 50 + 9.11 = 55.6, gain = 5.6; pre 100 -> post 0.93 * 100 + 9.1 = 102.1, gain = 2.1; pre 130 -> post 0.93 * 130 + 9.1 = 130.0, gain = 0.0).
3.3 Although the argument has been made that the effects of repeated measurement that are present in healthy adults are strongly reduced or even absent in children with epilepsy (Westerveld et al. 2000), this issue should definitely be addressed in the discussion, especially since other studies do suggest that practice effects cannot be ignored (Sherman et al. 2003).
3.4 The discussion repeatedly touches upon the pathogenesis of cognitive impairment (line 348; line 369; line 381). It seems to me that this, while not irrelevant, is beyond the proper scope of the manuscript, which provides no meaningful data on the pathogenesis of cognitive impairment beyond associations with particular etiologies.
3.5 The raw data are provided. However, to encourage their use, many small issues should be fixed:
- Column BN (“slopacienta“) repeats column A (patient_id). Please delete.
- Column K (seizure frequency): ‘Lesst_than_monthly’ should be ‘Less_than_monthly’.
- A few headers have spelling errors (e.g., AZ, BB, BD – ‘parcial’ should be ‘partial’). Please check and correct.
- The seizure classes listed in column N (1-4) appear to correspond to the seizure classes in columns AZ, BB, BC, and BD, respectively. The meaning of column N is unclear and duplicates the information in the other four columns. I would suggest deleting column N.
- Column Z is designated ‘MRS_finding’ but appears to specify only whether MRS was performed. I would suggest changing the header to simply ‘MRS’.
- Column AK: ‘Lesionektomy’ should be ‘Lesionectomy’.
- Column AO: the header ‘AWAKE’ in full caps suggests that this is an abbreviation, while it in fact appears to reflect simply whether awake surgery was performed.
- Column AS: some of the histopathological findings have not been (correctly) translated, e.g., ‘Jizva’ and ‘Encefalitida’. In addition, the header should read ‘histopathology’ rather than ‘histopatology’.
- Column AU: I would suggest applying the widely used Engel classification for seizure outcome (Engel et al. 1993), and coding this column accordingly. In addition, the ‘greater-than-or-equal-to’ symbol does not translate very well across computer platforms, so even if the current categories are maintained I would suggest changing the exact wording (e.g., ‘reductionover90pct’, since the percent sign suffers from the same problem though to a lesser degree).
- As far as I can tell the columns AT, BE-BI, BK, and BL all contain essentially the same information. I would suggest keeping only column AT.
- Similarly, column BM (‘epoch’) follows from column AH (although the exact cutoff, i.e., December 31st, 2010 vs January 1st, 2011 should be clarified in the manuscript itself).
- To reduce the file size (in terms of viewing, not so much in terms of disk space), I would also suggest removing columns BP and BQ since they duplicate the information in column AI.
- Why is the ‘surgery_extent’ (column AL) for some patients with the surgery type (column AK) ‘Hemisferectomy’ not ‘Hemisferectomy’ but rather ‘Focal resection’, ‘Multilobar resection’ or ‘Unilobar resection’?
- In column AK I would replace the label ‘Individual_resection’ with the more commonly used ‘Tailored_resection’, which also used in the text of the manuscript.

Additional comments

4.1 Given the use of multiple regression for the multivariate analysis, I would suggest framing the univariate analyses in terms of regression as well. While F-tests would still be required to assess the overall significance of effects that involve factors with more than two levels, for all other results a uniform method of reporting can then be used. Using the example of Table 1, each continuous variable or 2-level factor then only needs one line. For the univariate analyses, this line should report the variable, the intercept and its standard error, and the p-value (of the intercept). For the multivariate analyses, the global intercept should be reported on a separate line. The result could look something like this:
Determinants of pre-surgical IQ/DQ in whole dataset
Univariable regression Multivariable regression
Variable Intercept (SE) Beta (SE) / F p-value Beta (SE) / F p-value
Intercept - - - 89.55 (10.14) <0.001
Abnormal neurological finding1 86.83 (1.55) -18.46 (2.91) <0.001 -10.95 (3.47) 0.002
Epileptic syndrome2 64.08 (5.36) F2,188 = 6.83 0.001 F2,175 = 2.26 0.108
TLE 20.95 (5.73) < 0.001 9.53 (6.11) 0.120
XTLE 16.69 (5.73) 0.004 11.90 (5.70) 0.038
1 reference level: no abnormal neurological finding; 2 reference level: hemispheric epileptic
syndrome.
4.2 Custom checks:
Have you checked the authors ethical approval statement? Yes
Does the study meet our article requirements? Yes, but see 2.5
Has identifiable info been removed from all files? Yes
Were the experiments necessary and ethical? Yes
4.3 Raw data check – see 1.10 and 3.5
4.4 Image check – OK

Reviewer 2 ·

Basic reporting

No comment

Experimental design

No comment

Validity of the findings

No comment

Additional comments

This study is welcome and the authors have clearly a very good understanding of the field and of the issues. The sample size is large. I have a couple of small suggestions which the authors might like to consider when thinking about the study.

1. In the introduction the authors talk about seziure freedom in the context of epilepsy surgery outcome. I think while seizure freedom is a worthy goal signifcant seziure reduction (or disease improvement) is also worthy and this could be mentioned as for some patients seziure freedom may be difficult but seizure reduction is still a welcome result
2. The authors should mention as a limitation that different tests were used i.e. differnt versions of Wechsler and Bayley. I know this cannot be helped but it still makes comaprisons less valid due to use of IQ and DQ which are different concepts. There is also a risk of practice effects with only a 12 month period between restesting.
3. the authors might like to explain a little more why 2011 was chosen as the dividing line for the two time periods.
4. Lower IQ improves more- this might be discussed a little more - perhaps there is more room for improvement in low IQ etc.
5. In the results section i might be nice to state the direction of findings in some instances where it was not stated e.g there was a signifcant difference on post hoc analysis - group 1 was signifcantly better than group 2 etc.
6. In discussion perhaps discuss findings in comparison to styudies which ahve not found a signifcant improvment in cognitive scores e.g. VIGGEDAL, Gerd, et al. Intelligence two years after epilepsy surgery in children. Epilepsy & Behavior, 2013, 29.3: 565-570.

Minor points
please use . and not , when referring to decimal points

Recent - remove this as 2013 may not be recent for some people

In the first paragaph the term 'to rid' is used - please use a better term if possible

---

## Round 0.2 · Major Revisions

Dear Authors,

I believe your manuscript can be further improved by following the suggestions given by the peer-reviewer.

·

Basic reporting

The authors are to be commended for their substantial revision of the original manuscript. They have clearly put great effort into improving the manuscript, and they have sufficiently addressed the vast majority of the issues that I have previously raised. However, I believe there remain a few issues that merit further attention.

In my comments below, the numbering refers to my review of the first version of the manuscript, e.g., 1.6.1 is a comment regarding the issues discussed in point 1.6 of the previous review. Any comments that raise new issues are numbered in continuation of the previous numbering, e.g., 4.5.

1.6.1 The tables have much improved and are now easier to understand. However, I still feel that many tables contain too much empty space. For example, in the first part of Table 2 (Determinants of pre-surgical IQ/DQ), 49 of 88 cells (i.e., more than half) are empty. In this particular case:
- the second row can simply be deleted
- row 5 (epileptic syndrome), columns 2-4 should contain an estimate of effect size
- rows 8-9 (no/yes) can be deleted
- rows 12-last, columns 2-5 need to contain useful information or be deleted
I do not mean to imply that there should be no empty cells in the tables, but in their current form they take up more space than can be justified from their contents. Instead, some cells should probably be removed, and some cells need to be filled with information useful to the interested reader.
In addition, there are a number of inconsistencies to be addressed. To take the same example (Table 2):
- infantile spasms, status epilepticus and abnormal neurological finding are all factors with two levels, but only abnormal neurological finding gets three rows specifying the levels (two of which are almost empty – see above).
- epileptic syndrome, histopathology and FCD class are all factors with more than two levels, but only for FCD class are the levels specified in the table. If the reason for this is to clarify the precise classification used (e.g. FCD1/FCD2/FCD3 vs FCD1/FCD2A/FCD2B/FCD3A/FCD3B/unspecified MCD), then this should be done either in the table legend, or in the text of the manuscript. In the latter case the different classifications should be designated with labels which are then used throughout text and tables.
While I have here specified the problems specifically for one part of Table 2, these problems also apply to the rest of Table 2 and to the supplementary tables.

1.8.1 Commas still occur in lines 64 and 250. Additionally, in line 251 a period is missing (‘p=062’ [sic]), and in both line 250 and 251 there are commas missing between the reported numbers (‘F=0.48 p=0.49’ should be ‘F=0.48, p=0.49’).

1.9.1 The reporting of F-values still lacks indication of the degrees of freedom as well as effect sizes. The Publication Manual of the American Psychological Association (6th edition) gives the following example (p. 117):
‘For immediate recognition, the omnibus test of the main effect of sentence format was statistically significant, F(2, 177) = 6.30, p = .002, est ω2 = .07. The one-degree of freedom contrast of primary interest (the main difference between Conditions 1 and 2) was also statistically significant at the specified .05 level, t(177) = 3.51, p < .001, d = 0.65, 95% CI [0.35, 0.95].’
For many other results, only p¬-values are reported. I believe most of the corresponding information is reported in the revised (supplementary) tables, but if so it would be helpful to point out exactly where the information can be found. Where this is not the case (e.g., line 238 and probably other instances), the information should be reported in the text.

Experimental design

2.2.1 The rephrased research questions are a great improvement over the original version, and also correspond much better to the revised introduction. However, I think some further rephrasing may make it easier for the reader to grasp exactly what are the differences between the three questions. One source of confusion is the fact that the term ‘change’ [in cognitive performance] is used in two different meanings: individual change (post-surgical minus pre-surgical IQ) and historical changes (differences between the early and late period of the epilepsy surgery program). I would suggest something along these lines:
“In this study we aimed first to test whether there were differences in cognitive performance (pre-surgical IQ, post-surgical IQ, or change from pre-surgical to post-surgical IQ) between the early and late period of the paediatric epilepsy surgery program in our tertiary clinic. Next, we assessed whether there were differences in cognitive performance between the early and the late period for specific underlying aetiologies and epilepsy syndromes. Finally, we aimed to identify variables affecting cognitive performance in the entire cohort.”

2.9 I have belatedly realized that the authors reported using ‘generalized multiple regression’ (line 222). If this is correct, then the assumed distribution and the link function applied need to be specified (and some justification for these choices would be desirable). If, on the other hand, they used multiple regression based on a general (rather than a generalized) linear model, i.e., ordinary multiple regression, then this description should be corrected.

Validity of the findings

3.1.1 I have now realized that the reason that these values occur is because it is indeed the median that is being reported, and not, as I erroneously supposed when reading the first version, the mean. However, that raises the question why the median rather than the mean is reported: the authors use statistical inference techniques which rely on assumptions of normal distribution (and equal variances), but report the observed differences in terms that appear to suggest these assumptions are not met. Can the authors explain this choice?
On the other hand, Table 4 now reports post-hoc tests and reports a median difference which I suspect is instead a mean or estimated difference. In that case the column label should be corrected.

3.1.2 I have also realized that this is probably the reason why the authors don’t provide similar results for factors with more than two levels (e.g., FCD class in Table 2), since such a median difference (and Hodges-Lehmann confidence interval) cannot be provided for specific levels of the factor. However, I would still advocate for one of three options:
a) Reporting the results from the regression, including effects of specific levels through either treatment or sum contrasts (the contrasts used should of course be specified to allow interpretation – I think treatment contrasts are probably preferable because the interpretation is somewhat more straightforward). This is what I suggested in my previous review.
b) Reporting the pairwise median difference and its Hodges-Lehmann confidence interval in a manner similar to the treatment or sum contrasts mentioned in option 1. That is, for treatment contrasts one level (e.g., FCD1) would be set as the reference level, and the table would report the median difference between this level and each of the other levels. To approximate sum contrasts, one would report the median difference between each specific level and all other levels combined (e.g., FCD1 vs all others).
c) Reporting the median and its Hodges-Lehmann confidence interval for each level of the factor. This is conceptually different from the previous suggestions in that it does not provide an estimate of the difference in the technical sense (since the median difference does not necessarily equal the difference of the medians), but it does provide the reader with some feel for the pattern of factor level results.

3.3.1 However, I believe this [the issue of repeated measurement previously raised as point 3.3] is also relevant to the discussion regarding the greater IQ/DQ gain in children > 6 years (lines 318-322).

3.5.1
- A few headers have spelling errors (e.g., AZ, BB, BD – ‘parcial’ should be ‘partial’). Please check and correct.
-> The header for column BA should be ‘seizure_class_partial_complex’ rather than the current ‘seizure_class_partial_komplex’
- Column AU: I would suggest applying the widely used Engel classification for seizure outcome (Engel et al. 1993), and coding this column accordingly. In addition, the ‘greater-than-or-equal-to’ symbol does not translate very well across computer platforms, so even if the current categories are maintained I would suggest changing the exact wording (e.g., ‘reductionover90pct’, since the percent sign suffers from the same problem though to a lesser degree).
-> However, this classification should then also be used for the analyses (and reported in text and tables, e.g., Supplementary Table 3, lines 115-119).

Additional comments

4.5 The discussion has improved and is acceptable in its current form, but in my opinion it does not do justice to the rest of the paper. I would therefore highly recommend that the authors take another look at the discussion and attempt to condense it. There are many possible ways to do this and what follows are therefore really just suggestions.
For example, the structure of the discussion could be made to follow the structure of the analyses: overall comparison between the two periods, then the same comparison for subgroups (mainly for FCD, and then factors predicting cognitive performance in the sample as a whole. As another example, the first three paragraphs (lines 296-322) could be combined into a single paragraph discussing the interrelationships between age at surgery, pre-surgical IQ and IQ change after surgery. As a final example, some of the text is still phrased in ways that suggest that the issues discussed are addressed in the current study (e.g., lines 325-342) when in fact they are emphatically not analyzed here.
Again, I do not think that these changes are imperative for the manuscript to merit publication, but I do think that they would benefit the quality of the paper and – therefore – the reader.
4.6 Something appears to have gone wrong with the formatting of the references to the Wechler tests (lines 620-623) – please correct. I would also suggest providing the version information (1st edition / new revised edition) in English. Unless the journal policy on such matters prescribes differently, I would also suggest adding an English translation to publications not in English.
4.7 The reference for the Stanford-Binet scale (lines 608-609) provides a street address for the publisher. This is unusual, and also appears to be unnecessary.
4.8 There are (still) many minor formatting glitches. For example: in line 443 the text reads: ‘on-going(Boshuisen et al. 2015a).’ The correct spelling would be ‘ongoing’ (no hyphen), and a space is required between ‘ongoing’ and parentheses following it. I assume such issues will be addressed during copy editing, but they ought to be fixed before publication.
4.9 ‘epileptic sydrome’ [sic] appears twice in Table 2.

---

## Round 0.3 · Minor Revisions

Dear Authors,

Please revise for the minor repairs needed in your manuscript

·

Basic reporting

The authors deserve commendation for their persistence and hard work in revising the manuscript. The manuscript is now very nearly acceptable for publication. I have just a few small points that I do feel have to be addressed:
1.1 The authors still report that they performed **generalized** regression (line 225, line 281, the legend to Table 4, and Table 4 headers). **Generalized** regression implies that the distribution of either the dependent variables or the unobserved errors is assumed not to follow the normal distribution. If it is really true that they used a **generalized** regression analysis (and **not** a regular multiple regression based on the **general** linear model) then they do need to report details of this statistical model, including the distribution they used and the link function (quasi-normal with an identity link function?). However, the only thing in their results that I can see may suggest the use of a generalized regression is the dispersion estimate reported in Table 4. If this is not what they did then they should remove the word ‘generalized’ everywhere.
1.2 I appreciate the inclusion of the mean differences in Table 2. However, the authors should report a maximum of two decimals for these number (one decimal would be preferable, i.e., 9.0 or 9.03 instead of 9.03473). The authors should also clarify the meaning of the bracketed numbers in the columns ‘median difference’ and ‘mean difference’ – I think these are 95% confidence intervals? This should be explicit in the table itself, or in the table legend. In addition, the legend to Table 2 lists many abbreviations which are not used in the table.

Experimental design

no comment

Validity of the findings

no comment

Additional comments

no comment

---

## Round 0.4 · accepted · Accept

Congratulations! Your manuscript is its current form will be accepted and undergo galleyproof processing.

·

Basic reporting

no comment

Experimental design

no comment

Validity of the findings

no comment

Additional comments

I would recommend doing one more thorough spell check during the proofing stage to ensure that any errors that may have slipped in during the revisions (e.g., line 226 'modelby' instead of 'model by') are corrected.